# Microbial Regulation of Deterioration and Preservation of Salted Kelp under Different Temperature and Salinity Conditions

**DOI:** 10.3390/foods10081723

**Published:** 2021-07-26

**Authors:** Wei Wei, Xin Zhang, Zhaozhi Hou, Xinyu Hu, Yuan Wang, Caizheng Wang, Shujing Yang, Henglin Cui, Lin Zhu

**Affiliations:** 1School of Agricultural Engineering, Jiangsu University, Zhenjiang 212013, China; weiwei7096@ujs.edu.cn (W.W.); 2211916008@stmail.ujs.edu.cn (X.Z.); 2212016008@stmail.ujs.edu.cn (S.Y.); 2School of Food and Biological Engineering, Jiangsu University, Zhenjiang 212013, China; 2211818038@stmail.ujs.edu.cn (Z.H.); 2211816006@stmail.ujs.edu.cn (X.H.); 2211918033@stmail.ujs.edu.cn (Y.W.); 2222018071@stmail.ujs.edu.cn (C.W.); cuihenglin@ujs.edu.cn (H.C.)

**Keywords:** salted kelp, alginate lyase, halophilic archaea, metagenome

## Abstract

High salinity is an effective measure to preserve kelp, but salted kelp can still deteriorate after long-term preservation. In order to clarify the key conditions and microbial behavior of salted kelp preservation, 10% (S10), 20% (S20), and 30% (S30) salt concentrations were evaluated at 25 °C (T25) and 4 °C (T4). After 30 days storage, these salted kelps showed different states including rot (T25S10), softening (T25S20), and undamaged (other samples). By detecting polysaccharide lyase activity and performing high-throughput sequencing of the prokaryotic 16S rRNA sequence and metagenome, we found that deteriorated kelps (T25S10 and T25S20) had significantly higher alginate lyase activity and bacterial relative abundance than other undamaged samples. *Dyella*, *Saccharophagus*, *Halomonas*, *Aromatoleum*, *Ulvibacter*, *Rhodopirellula*, and *Microbulbifer* were annotated with genes encoding endonuclease-type alginate lyases, while *Bacillus* and *Thiobacillus* were annotated as the exonuclease type. Additionally, no alginate lyase activity was detected in undamaged kelps, whose dominant microorganisms were halophilic archaea without alginate lyase-encoding genes. These results indicated that room-temperature storage may promote salted kelp deterioration due to the secretion of bacterial alginate lyase, while ultra-high-salinity and low-temperature storage can inhibit bacterial alginate lyase and promote the growth of halophilic archaea without alginate lyase, thus achieving the preservation of salted kelp.

## 1. Introduction

Kelps, within the class Phaeophyceae, are subfrigid macroalgae endemic to the North Pacific; they are a very popular food in Asia due to their rich nutrients, delicious taste, and activities in preventing and treating chronic diseases, e.g., treating goiter and lowering blood sugar, blood pressure, and blood lipid levels [1,2,3,4]. In China, to prevent harvested kelp from deteriorating before putting it on the market, high-salinity (10–30%) kelp products are produced immediately under the processes of blanching, cooling, draining, and salt dewatering after receiving the kelp [5]. However, high salinity cannot completely prevent kelp from becoming locally red or rotten during its transportation or shelf-life [6]. According to incomplete statistics in China, the loss of salted kelp sold in supermarkets in the summer due to redness or rot reaches 10% to 30% of total sales, which represents tremendous economic losses to manufacturers and distributors.

Previous studies have shown that the microbial degradation of algal polysaccharides such as alginate, starch, and cellulose in the cell wall of kelp might lead to its deterioration [7]. Furthermore, it has been proven that some strains from bacterial genera such as *Klebsiella* [8], *Alteromonas* [9], *Pseudomonas* [10], *Sphingomonas* [11], *Flavobacterium* [2], *Saccharophagus* [12], *Pseudoalteromonas* [13], *Vibrio* [14], *Chitinophaga* [15], and *Defluviitalea* [16] can decompose algal polysaccharides, indicating that various bacterial groups can regulate the deterioration of kelp. However, if these bacterial groups decompose algal polysaccharides in the high-salt environment of the salted kelp surface, they have to exhibit salt tolerance. Only bacteria with salt tolerance and even halophilic physiological characteristics can complete the degradation of seaweed polysaccharides, which leads to the deterioration of kelp in salted kelp. To date, there have been no reports about the bacterial community dominating the degradation of kelp in salted kelp with high salinity. Additionally, some halophilic archaeal strains have been successfully isolated from stale kelp, and their characteristics of secreting red pigment and inducing kelp rotting have been described [17], leading researchers to suspect that halophilic archaea are the main cause of kelp decay; however, there is no research revealing whether and how halophilic archaea play a very important role in the deterioration of salted kelp. Therefore, identifying the community structure and function of prokaryotic microorganisms related to deterioration in salted kelp and determining the relevant preservation conditions that promote or inhibit these microbial communities will improve the understanding of the deterioration mechanism of salted kelp and provide a more targeted reference for the long-term preservation of salted kelp.

In this study, different temperatures and salinities were set up to preserve kelp. The states of the salted kelp samples after 30 days of preservation and the enzyme activities of three kinds of algal polysaccharide lyases, alginate lyase, amylase, and cellulase, were detected to determine the key conditions for the long-term preservation of salted kelp and the types of key biological enzymes that cause kelp to soften and rot. Subsequently, through high-throughput sequencing of microbial 16S rRNA amplicons and metagenomics, we clarified the dominant microbial communities and their sources in the salted kelp under different storage conditions and identified the microbial communities that can secrete key biological enzymes, as well as the types of key biological enzymes in rotted and undamaged salted kelp. These results will provide an important reference for analyzing the mechanism of rotting and maintenance of freshness of salted kelp and designing long-distance transportation and improved shelf-life measures, as well as provide information for the excavation of the microbial resources for degrading high-salt kelp waste.

## 2. Materials and Methods

### 2.1. Kelp Collection and Setup

Fresh kelp was collected in June 2018 in Rongcheng, Shandong Province, the largest kelp-producing area in China. The kelp was harvested during the day. Then, in accordance with the conventional preservation steps of the local kelp production factory, under the aseptic conditions of ultraviolet light, the kelp samples were cleaned using sterile water, blanched (hot water at 90 °C), cooled, and hung to drain at room temperature. Lastly, crude sea salt from the local salt field in Rongcheng was used for kelp salting at different salt concentrations.

The kelp samples were divided into two batches to set the salinity gradient. For each batch, crude sea salt and drained kelp samples were mixed at ratio of 10%, 20%, and 30% of the weight, and mixed to obtain three salinity gradients. One batch of samples was stored at 25 °C and labeled T25S10, T25S20, and T25S30. The other batch was stored at 4 °C and labeled T4S10, T4S20, and T4S30. At the same time, the crude sea salt for kelp salting was configured as a saturated solution as a blank control and labeled S. Each sample group had three sets of parallel samples as experimental repeats and was stored in incubators at 4 and 25 °C with a room humidity (RH) of 50% in dust-free conditions for 30 days. During the 30 day preservation period, the status of these salted kelp samples was monitored at all times. At the end of the preservation, the salted kelp samples of each group were homogenized using a cell crushing apparatus for subsequent research after the salt particles were removed.

### 2.2. Detection of Bioenzyme Activity in Salted Kelp Samples

The activities of amylase, cellulase, and alginate lyase in the abovementioned salted kelp samples and saturated salt solutions under different preservation conditions were detected. Amylase activity was measured according to the PTK method [18]. The cellulase enzyme activity was determined using an electrochemical sensor described in a previous study [19]. The enzyme activity of alginate lyase was determined by the thiobarbituric acid method [20].

### 2.3. High-Throughput Sequencing of Amplicons and Metagenomics in Salted Kelp Samples

Genomic DNA was extracted from the salted kelp samples and saturated salt solution. The prepared salted kelp homogenate was subjected to high-speed centrifugation at 4 °C and 17,000 rpm for 10 min to obtain a precipitate. One gram of the precipitate was collected, and microbial genomic DNA extraction was performed according to the instructions of the PowerSoil^®^ DNA Isolation Kit. Then, 500 mL of saturated saline solution was used to extract the microbial genomic DNA according to the instructions of the PowerWater^®^ DNA Isolation Kit.

The extracted microbial genomic DNA was sent to Beijing Nuohe Zhiyuan Technology Company for high-throughput sequencing. The Illumina-HiSeq sequencing platform was used in combination with the universal primers U515F and U806R for the 16S rRNA gene of prokaryotic microorganisms to perform high-throughput sequencing of PCR amplicons of all the samples. Then, the Illumina-PE150 sequencing platform was used to fragment the genomic DNA of the selected rotten (T25S10) and non-rotten (T4S10) salted kelp into 350 bp fragments for metagenomic sequencing.

### 2.4. High-Throughput Sequencing Data Analysis

The raw amplicon sequencing data were optimized using Trimmomatic and FLASH, and the sequences were clustered according to 97% similarity into operational taxonomic units (OTUs). During the clustering process, UCHIME chimaeras were removed, the results were compared using the Silva database, and relevant algorithms were used for subsequent analysis. Principal component analysis (PCA) is a statistic method which reduces the dimensionality of large datasets without losing statistic information. The OTUs in different samples were analyzed by PCA using the OTU abundances through Originlab software.

The raw metagenomic sequencing data were subjected to quality control to obtain clean data, which were assembled in Metagenome. MetaGeneMark was used for gene prediction and gene catalog construction. On the basis of the gene catalog, the Carbohydrate-Active Enzymes (CAZy) database was used to perform functional annotation and abundance analysis of carbohydrate enzymes, including alginate lyase, proteolytic enzymes, and cellulase.

## 3. Results

### 3.1. The States of Salted Kelp Samples after 30 Days of Preservation

A 30 day continuous monitoring showed that the stored kelp samples showed three different sensory states, including softening, rotten, and undamaged. Softening is a sensory state between rotten and undamaged, which means that the surface of the kelp loses its luster and is easily broken under external pressure. The T25S10 sample began to soften on the 13th day and rotted to a great extent by 30 days. The T25S20 sample also began softening on the 30th day, while the T25S30 sample and all the kelp samples at 4 °C were undamaged during this period (Figure 1).

### 3.2. Activity of Algal Polysaccharide Lyase in Salted Kelp Samples

The activities of three kinds of algal polysaccharide lyases in the salted kelp samples are shown in Figure 2. No alginate lyase activity was detected in the three salt gradient samples stored at 4 °C; however, the alginate lyase activity of the T25S10 and T25S20 samples was significantly higher than that of the kelp samples under other storage conditions (*p* < 0.05), and the T25S10 sample showed a significantly higher alginate lyase activity than that of any other sample. At the same time, both amylase and cellulase enzyme activities were detected, but there was no significant difference in all the salted kelp samples (*p* > 0.05). Combined with the results from Figure 1, these results indicated that the alginate degradation process dominated by alginate lyase may be the main cause of the softening and decay of salted kelp, and low-temperature storage may be an effective method to inhibit alginate lyase activity.

### 3.3. Microbial Community Diversity in Salted Kelp Samples

To analyze the diversity and source of the microbial community in these kelp samples preserved under different conditions, high-throughput sequencing libraries of prokaryotic 16S rRNA for all salted kelp samples, as well as the saturated salt solution sample, were constructed. The coverage of all sequencing libraries was over 99.5% (Table 1), indicating that all the high-throughput sequencing libraries had sufficient sequencing depth. The rotten sample T25S10 had 554 OTUs (Table 1), which was the highest among all the samples, and the number of OTUs specific to that sample was as high as 177. Furthermore, the T25S10 sample had the highest Chao1 index value and the lowest Shannon and Simpson indices. The Shannon diversity and Simpson index of T25S30, T4S10, T4S20, T4S30, and S (saturated solution) samples showed no differences (Table 1). The results of principal component analysis (PCA) based on OTUs showed that the PCA scores of the bacterial and archaeal communities from the softened sample T25S20 and rotten sample T25S10 were similar and were significantly different from those of the T25S30 sample, all the kelp samples at 4 °C (T4S10, T4S20, T4S30), and the saturated solution (S) (Figure 3). These results indicate that the bacteria and archaea in the rotten kelp sample formed a more diverse and specific prokaryotic microbial community structure than the undamaged salted kelp samples during the same preservation period.

### 3.4. Composition and Source of Microbial Communities in Salted Kelp Samples

The annotation results at the phylum level are shown in Figure 4. The dominant phyla in the T25S10 sample were Firmicutes (relative abundance 50.1%), Proteobacteria (14.7%), and Euryarchaeota (32.8%). With regard to the T25S20 sample, Euryarchaeota was the most dominant phylum (82.8%), with Firmicutes and Proteobacteria accounting for only 7.9% and 2.5% of the phyla, respectively. For all other samples, including all samples stored at 4 °C and in a 30% salt concentration at 25 °C, the microbial communities were mainly archaea, including Euryarchaeota and Nanohaloarchaeota, whose relative abundances were between 91.0% and 98.9%. The relative abundance of the bacterial community was low (1–9%). The results indicate that Firmicutes and Proteobacteria may play a crucial role in the softening and rotting of salted kelp, and that the archaea in Euryarchaeota and Nanohaloarchaeota may play an important role in the long-term preservation of salted kelp.

The top four microbial phyla in the saturated salt solution sample (S) were the most dominant group in the rotten and undamaged salted kelp samples mentioned above, namely, Euryarchaeota (77.3%), Nanohaloarchaeota (20.7%), Firmicutes (1.2%), and Proteobacteria (0.7%) (Figure 4). The results indicate that the dominant bacterial or archaeal communities in the rotten and undamaged salted kelp samples were likely derived from the crude sea salt, which formed different community compositions under different preservation conditions.

### 3.5. Microbial Function Analysis of Rotten and Undamaged Salted Kelp Based on the Metagenome

The rotten sample T25S10 and undamaged sample T4S10 were selected to perform microbial function analysis using metagenomic sequencing technology. Metabonomic sequencing of 1 g of the T25S10 and T4S10 kelp samples yielded 371 Mb and 225 Mb, respectively. After splicing, the number of effective functional genes was 3.9 × 10^5^ and 2.2 × 10^5^, respectively (Table 2). For the total genomic DNA obtained from the sample T25S10, 77% of the gene sequences were annotated to be from bacteria, 3% were from archaea, and 19% were unknown microbial sequences that could not be annotated. With regard to the sample T4S10, 88% of the gene sequences were annotated to be from archaea, 4% were from bacteria, and only 8% of the gene sequences were unknown sequences (Table 2). These results indicated that bacteria and archaea dominated the microbial community of deteriorated and undamaged salted kelp samples, respectively.

The CAZy database divides the annotated enzyme genes into six major carbohydrate-related categories based on the similarity of amino acid sequences in the enzyme protein domain, namely, glycoside hydrolases (GHs), glycosyltransferases (GTs), polysaccharide lyases (PLs), carbohydrate esterases (CEs), auxiliary oxidoreductases (AAs), and carbohydrate-binding modules (CBMs). The T25S10 and T4S10 samples were annotated with the CAZy database to obtain 18,712 and 4634 carbohydrate metabolism-related enzyme genes, respectively. Obviously, the T25S10 sample had a higher number of the above six types of enzyme genes than the T4S10 sample. According to the results of existing studies, alginate lyases were mainly distributed in the PL5, PL6, PL7, PL14, PL15, PL17, PL18, PL36, and PL39 families among the 41 PL families (Table 3). The rotten sample T25S10 exhibited 654 genes encoding PL (Table 2), and 87 of them encoded alginate lyases, which belonged to the PL5, PL6, PL7, PL14, PL15, and PL17 families (Figure 5). However, only 56 PL genes were obtained from the undamaged sample T4S10 (Table 2), and none of them encoded alginate lyase. This result is in agreement with the above enzyme activity test results, showing that, at 25 °C, a 10% salt concentration could promote the secretion of various alginate-lysing enzymes by microbial communities in salted kelp, while, at 4 °C, secretion was completely inhibited.

### 3.6. Species Annotation of Rotten and Undamaged Salted Kelp Microbial Metagenomes

After annotation, it was found that the 87 alginate lyases belonging to PL5, PL6, PL7, PL14, PL15, and PL17 in the T25S10 sample were derived from nine known and three unknown genera in five bacterial phyla. Among these genera, 7% were *Ulvibacter* derived from Bacteroidetes, 7% were *Rhodopirellula* from Planctomycetes, 3% were an unknown genus of Actinobacteria, 10% were *Bacillus* from Firmicutes, and 72% were derived from Proteobacteria, including *Dyella* (21%), *Saccharophagus* (14%), *Halomonas* (12%), *Aromatoleum* (7%), *Microbulbifer* (3%), *Thiobacillus* (2%), and two unknown genera (14%). These bacterial groups with alginate lyase genes were different from the bacterial sources reported previously (Table 3). Additionally, among the alginate lyases produced by these microorganisms, endo-type alginate lyases of PL6, PL7, and PL14 and exo-type alginate lyases of PL17 were the most dominant types, accounting for 45%, 14%, 28%, and 10% of the total, respectively (Figure 5). These results indicated that the various alginate lyases in the rotten salted kelp samples were derived from a variety of bacterial genera, which may gradually complete the softening and decay of the salted kelp through combined action.

The microbial species annotation results of the T4S10 sample showed that 70% of the sequences were derived from Halobacteria (Figure 6). Among these halophilic archaeal groups, only 50% of the total number could be annotated, and they belonged to 20 genera in the Halobacteria class, which indicated the diversity of halophilic archaeal groups in undamaged kelp samples. However, no alginate lyase gene was found in the genomes of these dominant and diverse halophilic archaea. Although a small proportion of bacterial communities were noted, none of them were the above 12 bacterial genera with alginate lyase, and no other genes encoding alginate lyase were found. These results indicated that the halophilic archaea in salted kelp samples stored in ultra-high salt concentration or low temperature might replace the bacterial community that can secrete alginate lyase, thereby effectively preventing the action of alginate lyase on kelp decomposition.

## 4. Discussion

We monitored the salted kelp samples under different storage conditions for 30 consecutive days, and the results showed that, at 25 °C, even when the salinity was as high as 10% or 20%, the kelp still became soft or even decayed, causing deterioration during storage. At the same time, we detected significant alginate lyase activity in these spoiled salted kelp samples. However, when the salt concentration was 30% or the temperature was 4 °C, the preserved salted kelp samples showed the same status as fresh kelp, and no alginate lyase activity was detected. These results indicate that the degradation of alginate is a determinant of the softening and decay process of salted kelp. In addition, room temperature could promote the degradation process of the salted kelp and lead to its decay and deterioration, while ultra-high-salinity or low-temperature preservation could inhibit the degradation of alginate to prevent the salted kelp from degrading. Furthermore, two high-throughput sequencing results showed that the bacterial community derived from crude sea salt provided the dominant microorganisms in the metamorphic kelp samples. It was found that some bacteria belonging to Proteobacteria [27], Actinobacteria [28], Bacteroidetes [29], and Firmicutes [16] contained genes that can encode a variety of alginate lyases, which may be the reason why these microorganisms caused the salted kelp to soften and deteriorate. In contrast, for the undamaged kelp samples, the dominant microorganisms were the archaeal community derived from crude sea salt. Among them, halophilic archaea that could not secrete any types of alginate lyase exhibited a significant advantage. These halophilic microorganisms are not influenced by ultra-high salt concentrations [30] and, therefore, pre-emptively occupied the limited niche in extremely highly salted kelp, which indirectly inhibited the growth and reproduction of related bacterial communities that could cause the deterioration of salted kelp. Therefore, during the transportation and shelf-life of kelp, measures such as cold chain transportation, cryogenic storage, or ultra-high salt concentration treatment can effectively inhibit the growth and reproduction of microorganisms capable of secreting alginate lyase, thereby extending the shelf-life of kelp.

Alginate is a polysaccharide irregularly connected by α-1,4-d-mannuronic acid and β-1,4-l-guluronic acid, which accounts for approximately 20% of kelp [31]. Alginate is an important part of the kelp cell wall [32]. Therefore, the degradation of alginate by microbial alginate lyase can cause the kelp to soften or even rot. Under high-salt conditions, the activity of microbial alginate lyase can be inhibited for two reasons: the inhibition of microbial growth and metabolism and inhibition of the activity of alginate lyase secreted by microorganisms. With regard to the former reason, the high-throughput result of this study verified that alginate lyase-secreting bacteria could be inhibited by high salt concentration, which could indirectly impact the activity of alginate lyase in the tested kelp samples. With regard to the latter reason, some reports on salt-tolerant microorganisms can provide corresponding support. An alginate lyase from marine *Vibrio* sp. NJ-04 with high salt tolerance was activated by NaCl in the range of 200–700 mM, while the enzyme activity was inhibited beyond 700 mM [33]. Thus, this microbial alginate lyase is sensitive to salt concentration.

It has been shown that alginate lyases are of various types, mainly belonging to the PL5 [21], PL6 [22], PL7 [23], PL8 [14], PL14 [24,25], PL15 [11], PL17 [26], PL18 [13], PL36 [15], and PL39 [16] families. Alginate lyases can be divided into endo-type and exo-type according to their degradation method [34]. Endo-type alginate lyases can degrade alginate into alginate oligosaccharides [35], while exo-type alginate lyases can degrade alginate oligosaccharides into monosaccharides [36,37,38]. The metagenomic sequencing and annotation results indicated that 88% of the total annotated alginate lyases were endo-type belonging to the PL5, PL6, PL7, and PL14 families, while the remainder were exo-type belonging to the PL15 and PL17 families. Therefore, endo-type alginate lyases secreted by salt-tolerant bacteria derived from *Dyella*, *Saccharophagus*, *Halomonas*, *Aromatoleum*, *Ulvibacter*, *Rhodopirellula*, and *Microbulbifer* may cause salted kelp to soften by degrading alginate to alginate oligosaccharides. Then, exo-type alginate lyases secreted by salt-tolerant bacteria, such as *Bacillus* and *Thiobacillus*, can further degrade alginate oligosaccharides into monosaccharides, eventually causing the decay of salted kelp.

Although the abovementioned salt-tolerant bacteria can seriously endanger the kelp industry by secreting alginate lyase, they are also beneficial to humans. For example, due to the generally low utilization rate of algal polysaccharides, after extracting the active ingredients, the remaining algal waste still contains a large amount of alginate, whose degradation products still have high biological value, such as antiviral and antibacterial activities [2]. Therefore, the endo-type alginate lyase-secreting bacteria identified in this study can be used as a source of salt-tolerant strains or salt-tolerant biological enzymes for alginate degradation, which would provide a powerful microbial tool for degrading alginate in algal waste into alginate oligosaccharides with high added value.

This study also revealed a large number of halophilic archaea in fresh kelp samples. Studies have shown that these microorganisms can counter the hyperosmotic environment outside the cell by adjusting the osmotic pressure inside and outside the cell such that it has a certain tolerance to a high-salt environment [39], potentially even growing normally in a 30% salt concentration [40]. However, it is very difficult for them to adapt to the osmotic pressure in the normal human body. Even washing with water can cause rupture and death of these organisms [41]; thus, it is nearly impossible for them to survive in the human body. Although there is no evidence so far that these halophilic archaea can inhibit the growth of bacteria that have the ability to secrete alginate lyase, they can multiply normally in salted kelp and seize the limited niche of salted kelp, thereby inhibiting the growth and reproduction of other bacterial groups. Therefore, this unique microbial ecological phenomenon may also be a new biological preservation method to maintain the freshness of salted kelp.

## 5. Conclusions

In this study, by observing and studying salted kelp preserved at different temperatures and salt concentrations, we found that salted kelp softened and even rotted under the action of alginate lyases at room-temperature storage with salinities below 20%. In this case, the microbial community on the salted kelp was dominated by a variety of edible salt-derived bacterial communities, which could secrete various endo-type and exo-type alginate lyases, and the combined action of these enzymes led to salted kelp softening and rotting. However, the kelp remained healthy, and no alginate lyase was detected when it was preserved at room temperature in ultra-high-salinity conditions or at low temperature with salinity between 10% and 30%. In this case, the dominant microorganisms on the salted kelp were halophilic archaea, which could not secrete alginate lyase, thus forming a storage condition that would help the salted kelp not deteriorate.

## Figures and Tables

**Figure 1 foods-10-01723-f001:**
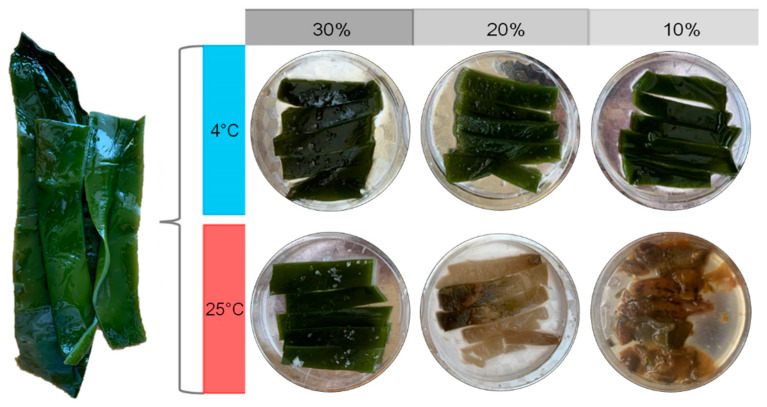
States of fresh salted kelp and salted kelp with different salt concentrations at 4 °C and 25 °C after 30 days of preservation.

**Figure 2 foods-10-01723-f002:**
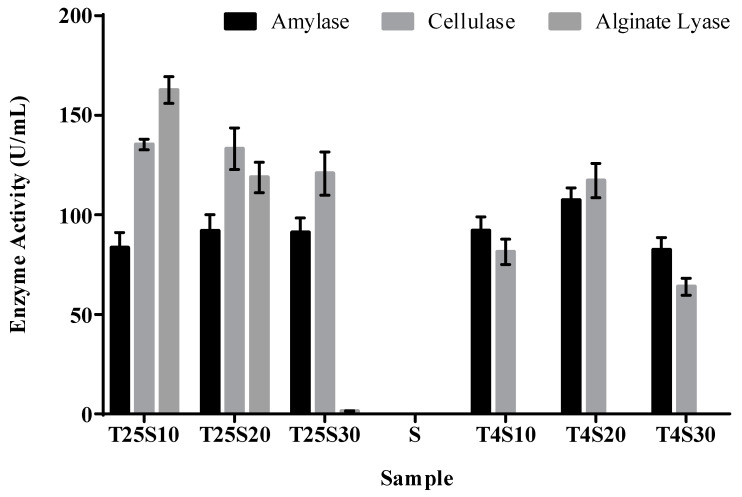
Enzyme activity determination results of amylase, cellulase, and alginate lyase in different treatment salted kelp samples.

**Figure 3 foods-10-01723-f003:**
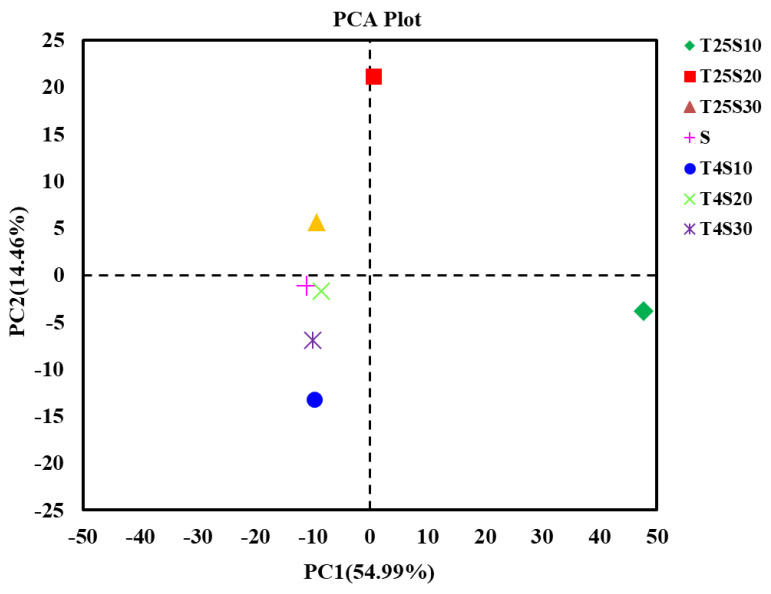
The principal component analysis (PCA) results of the salted kelp samples under different preservation conditions.

**Figure 4 foods-10-01723-f004:**
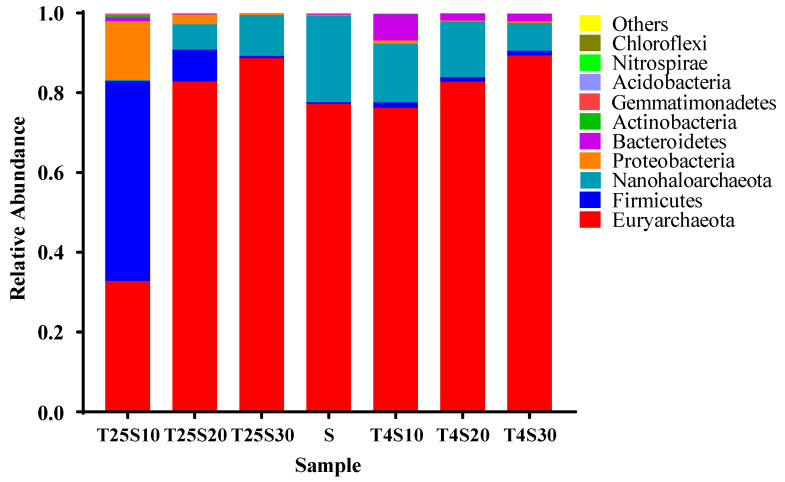
The relative abundance of the top 10 microorganisms at the phylum level in different treatment salted kelp samples.

**Figure 5 foods-10-01723-f005:**
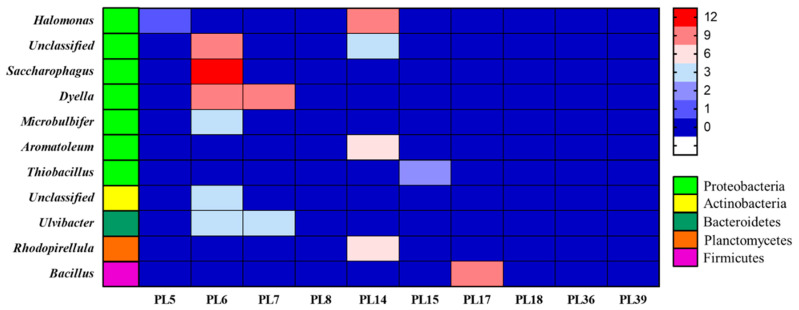
The microbial sources of 87 alginate lyases in T25S10 sample.

**Figure 6 foods-10-01723-f006:**
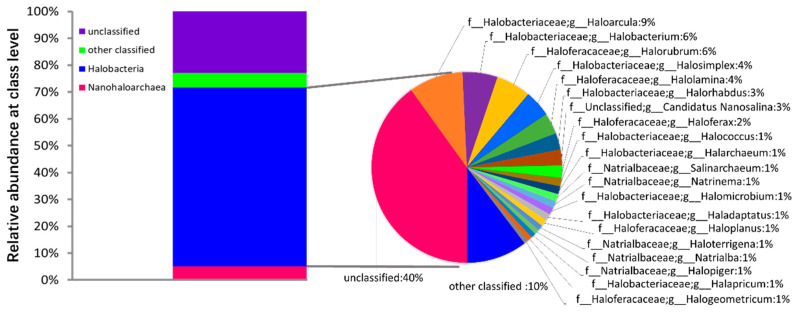
The microbial species annotation results of T4S10 sample.

**Table 1 foods-10-01723-t001:** High-throughput sequencing library information of bacterial and archaea 16S rRNA sequences from the salted kelp samples under different preservation conditions.

Sample	OTU	Unique OTU	Shannon Index	Simpson Index	Chao1	Goods_Coverage
T25S10	554	177	3.58	0.77	699.10	99.6%
T25S20	331	12	4.35	0.88	393.58	99.7%
T25S30	233	4	4.91	0.91	268.00	99.9%
S	287	15	5.63	0.96	346.43	99.8%
T4S10	305	9	5.63	0.96	401.28	99.8%
T4S20	334	4	5.88	0.97	469.30	99.7%
T4S30	296	8	5.49	0.95	445.50	99.7%

Note: OTU, operational taxonomic units.

**Table 2 foods-10-01723-t002:** Library information of microbial metagenome sequencing of rotten (T25S10) and unrotten (T4S10) salted kelp samples.

Item	T25S10	T4S10
Total Len. (bp)	389,089,268	236,304,057
Max Len. (bp)	43,046	94,615
Average Len. (bp)	998.89	1069.79
N50 Len. (bp)	1023	1128
N90 Len. (bp)	558	571
Total Num.	389,522	220,889
Archaeal Num.	12,231	193,350
Bacterial Num.	301,263	9161
Eukaryota Num.	344	144
Virus Num.	948	722
Others	74,736	17,512
Total Num. of Carbohydrate-Active enzymes	18,712	4634
PL	654 (87)	56 (0)
GT	5641	2504
GH	6247	1393
CE	2527	98
CBM	3420	511
AA	223	72

**Table 3 foods-10-01723-t003:** Summary of different types of alginate lyases in the PL families.

PL Family	Enzyme	Action Mode	Source of First Report	Environment for Sample Isolation	References
5	AlyA1-III	Endo-	*Pseudomonas* sp. E03	Mud	[21]
6	AlyMG	Endo-	*Stenotrophomonas* *maltophilia* KJ-2	-	[22]
7	Pa1167	Endo-	*Pseudomonas aeruginosa* PAO1	Coastal marine habitats	[23]
8	Vpa_0049	Endo-	*Vibrio* sp. QY108	Marine	[14]
14	vAL-1	Endo-	*Halitos discus hannai*	Abalone	[24]
			*Chlorella virus* ATCV-1	Heliozoon *A. turfacea*	[25]
15	Atu3025	Exo-	*Sphingomonas* sp. A1	Soil	[11]
17	Alg17C	Exo-	*Saccharophagus degradans*	Marine	[26]
18	Aly-SJ02	Endo-	*Pseudoalteromonas* sp. SM0524	Marine rotten kelp	[13]
36	Aly36B	Endo-	*Chitinophaga* sp. MD30	Air conditioner condensate pipe	[15]
39	Dp0100	Endo-	*Defluviitalea phaphyphila*	Coastal sediment	[16]

## Data Availability

The data presented in this study are available on request from the corresponding author.

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
