# Peer review of "Microbial Regulation of Deterioration and Preservation of Salted Kelp under Different Temperature and Salinity Conditions"

_foods, 2021, doi:10.3390/foods10081723_

Round 1

Reviewer 1 Report

Dear Authors,

except for some corrections (see attached file), the manuscript entitled "Microbial regulation of deterioration and preservation of salted kelp under different temperature and salinity conditions" is well written, simple and effective. In my opinion it is an interesting research .

Author Response

Point 1: except for some corrections (see attached file), the manuscript entitled "Microbial regulation of deterioration and preservation of salted kelp under different temperature and salinity conditions" is well written, simple and effective. In my opinion it is an interesting research .

Response 1: Thank you for noticing the errors. We have revised the microorganism names that need to be italicized and the journal names that need to be abbreviated.

Reviewer 2 Report

Foods (MDPI): Microbial regulation of deterioration and preservation of slated kelp under different temperature and Salinity conditions.

Comments and suggestions:

This paper is well-written and contains well-supported interesting and useful information on rotting kelp and its potential preservation. Only clarifications and suggestions are necessary.

  1. In Section 2.1 a bit more detail should be provided to facilitate repetition by others. Details such as draining method and centrifugation details should be included. The definition of salinity is unclear because it appears based on sample weight (dry/wet?) rather than solution concentration, which would be preferred. Provide details on storage conditions including humidity/dust free…etc.
  2. Section 3, lines 132-135 contain author advice and should be removed.
  3. Section 3.1 should include some clarifying statement on what constitutes softening. A clear definition would be helpful.
  4. In Section 3.2 (Fig 2) it is recommended that some comments discussing the effect of salt concentration on alginate activity (at 25oC) be included here.
  5. Line 169-172. This sentence is too long and should be split.
  6. Section 3.3, Figure 3. It is currently a bit unclear how the PCA analysis was set up. Additional details and explanation and interpretation, including a reference to the analysis software, would help.    
  7. Line 206-208 and 268-271 and 292-293. Comments regarding microbial consortia being derived from the crude sea salt are a bit subjective. Other options for introduction of those bacteria should be tested and discussed prior to exclusion (fresh kelp prior to storage, for example?). Alternatively this discussion and conclusions should be cut.
  8. Figure 6. The text on the RHS of the figure is too small…revise for enhanced clarity. The use of the term healthy fresh kelp samples here is confusing….was it one of your samples which showed reduced rotting (T25S30, for example) or was it prior to storage (in which case the sample should be clearly defined).

Author Response

Point 1: In Section 2.1 a bit more detail should be provided to facilitate repetition by others. Details such as draining method and centrifugation details should be included. The definition of salinity is unclear because it appears based on sample weight (dry/wet?) rather than solution concentration, which would be preferred. Provide details on storage conditions including humidity/dust free…etc.

Response 1: Thank you for your suggestion. We added more details about the methods and storage conditions in lines 79-85 and lines 90-91.

Point 2: Section 3, lines 132-135 contain author advice and should be removed.

Response 2: Thank you for your suggestion. We have removed the sentences.

Point 3: Section 3.1 should include some clarifying statement on what constitutes softening. A clear definition would be helpful.

Response 3: Thank you for your suggestion. We agree with your comments and have added a clear definition for softening in lines 130-131.

Point 4: In Section 3.2 (Fig 2) it is recommended that some comments discussing the effect of salt concentration on alginate activity (at 25oC) be included here.

Response 4: Thank you for your suggestion. We have added the discussion about the effect of salt concentration on alginate lyse activity in lines 295-300.

Point 5: Line 169-172. This sentence is too long and should be split.

Response 5: Thank you for your suggestion. We have split and paraphrased this sentence.

Point 6: Section 3.3, Figure 3. It is currently a bit unclear how the PCA analysis was set up. Additional details and explanation and interpretation, including a reference to the analysis software, would help.

Response 6: Thank you for your suggestion. We revised the section 3.3 and added the details about PCA analysis into the section 2.4.

Point 7: Line 206-208 and 268-271 and 292-293. Comments regarding microbial consortia being derived from the crude sea salt are a bit subjective. Other options for introduction of those bacteria should be tested and discussed prior to exclusion (fresh kelp prior to storage, for example?). Alternatively, this discussion and conclusions should be cut.

Response 7: Thank you for your suggestion. In fact, the surfaces of kelp samples were disinfected before setup. Thus, combined with the results of high-throughput sequencing, we can indirectly infer that the microorganisms in salted kelp samples may come from edible salt. Considering your opinion and the possibility of over discussion, we decided to remove the corresponding description from the sections of abstract and conclusion, and only mention it in the discussion section.

Point 8: Figure 6. The text on the RHS of the figure is too small…revise for enhanced clarity. The use of the term healthy fresh kelp samples here is confusing….was it one of your samples which showed reduced rotting (T25S30, for example) or was it prior to storage (in which case the sample should be clearly defined).

Response 8: Thank you for your suggestion. We have revised the Fig. 6, and used the word " undamaged kelp " to replace the word "healthy kelp" and "fresh kelp" throughout the manuscript.